# A Cell-Free DNA Plasma Next-Generation Sequencing Test—Is It Worth the Cost?

**DOI:** 10.3390/pathogens14080811

**Published:** 2025-08-15

**Authors:** Sean Jung, Francesca Torriani, Shira Abeles, Ahnika Kline

**Affiliations:** 1Department of Medicine, Division of Infectious Diseases and Global Public Health, University of California San Diego, La Jolla, CA 92093, USA; ftorriani@health.ucsd.edu (F.T.); sabeles@health.ucsd.edu (S.A.); 2Infection Prevention and Clinical Epidemiology and Antimicrobial Stewardship Programs, University of California San Diego Health, San Diego, CA 92103, USA; 3Department of Pathology, Center for Advanced Laboratory Medicine, University of California San Diego, La Jolla, CA 92093, USA

**Keywords:** cell-free DNA, Karius Test, next-generation sequencing, infectious disease diagnostics, cost-effectiveness, clinical utility, stewardship

## Abstract

Background: The Karius Test (KT), a microbial cell-free DNA next-generation sequencing assay, is increasingly utilized in challenging infectious syndromes. However, its real-world clinical utility and cost-effectiveness remain uncertain. Methods: We conducted a retrospective review of 88 KT results from adult patients at UC San Diego Health between July 2017 and April 2024. Each case was evaluated for clinical impact using standardized criteria. We analyzed diagnostic yield, turnaround time, number needed to test (NNT), and institutional billing data for reimbursement and cost implications. Results: Of 88 unique tests, forty-nine (55.7%) identified at least one pathogen. Eleven (12.5%) had a positive clinical impact—eight provided the only microbiologic diagnosis, and three were faster than conventional methods. Vascular/graft infections showed the highest yield. Twenty-one tests had a neutral impact; fifty-six showed no clinical benefit. The Median turnaround time was 3 days. The NNT was 6.1 or 2.75 including neutral cases. Cost analysis revealed a substantial financial burden without transparent reimbursement mechanisms in inpatient settings. Conclusion: The KT demonstrates modest clinical utility with noteworthy benefits in select scenarios. Given its high cost and variable impact, we advocate for diagnostic stewardship led by infectious disease specialists to optimize test use and minimize unnecessary expense.

## 1. Introduction

Approximately 1400 microorganisms are known to cause human disease [1], resulting in more than 10 million outpatient visits and 482,000 hospital admissions in 2019, in the United States [2]. Early pathogen identification is critical to targeting effective therapy and fewer adverse events, but identification of the causative pathogen remains challenging in many clinical scenarios despite advancements in molecular diagnostics. With the refinement of genomic medicine, we have witnessed transformative advancements in next-generation sequencing (NGS) technologies. This advancement is having a profound impact in all areas of medicine, particularly in the diagnosis of infectious diseases, when traditional methods often fail to identify organisms [3].

In the rapidly evolving field of genomics and diagnostics, various NGS technologies offer distinct advantages for analyzing complex biological samples. There are three distinct approaches when applying NGS technologies to diagnose infectious diseases: whole-genome sequencing, targeted NGS, and metagenomic NGS. Each method provides unique insights into microbial communities and infectious diseases, and understanding their differences can guide clinicians in selecting the most appropriate diagnostic tool [4]. However, that is beyond the scope of this brief article.

Sequencing of cell-free DNA (cfDNA) has recently been shown to have clinical utility, especially in infections among pediatric populations, immunocompromised patients with infections caused by fastidious or slow-growing organisms like fungi or mycobacteria, or for molecular diagnostics of deep-seated infections that would otherwise require invasive procedures for diagnosis [5,6,7]. Clinical evidence is building, with more than 300 articles available as clinical data, though the results from randomized clinical trials are scarce [8]. The Karius cfDNA NGS Test (KT) represents a distinct approach that analyzes cell-free DNA fragments circulating in the bloodstream. This method has unique features and applications, including non-invasiveness and broad pathogen detection [9]. However, the KT may face challenges related to the quality of the sample, the potential for false positives or negatives, the lack of utility for RNA viruses, and high cost.

In this study, we aimed to evaluate the real-world utilization, clinical outcomes, and cost implications of the KT at our institution as a quality improvement initiative. Prior to our evaluation, the test was made available for ordering by any provider without infectious disease consultation, solely under the guidance of microbiology laboratory leadership. We sought to understand how frequently and for what indications it was being used, assess its diagnostic and clinical utility, and determine whether its use was associated with beneficial or adverse outcomes. Importantly, we also examined the cost-effectiveness of the test in routine clinical practice. To our knowledge, few retrospective studies have systematically analyzed institutional-level data on KT utilization across a broad range of patients and specialties, without focusing on the cost-effectiveness; this highlights a critical gap in the literature that this analysis seeks to address.

## 2. Methods

When a KT is ordered at our institution, a blood sample (approximately 5 mL) is collected during routine care in EDTA vacutainer plasma preparation tubes. Samples are sent to the lab, centrifuged to obtain plasma, then shipped overnight to Karius. Karius performs cfDNA extraction upon receiving the samples, and next-generation sequencing libraries are prepared using customized dual-indexed Ovation^®^ Ultralow System V2 library preparation kits (Tecan, Männedorf, Switzerland) in STAR™ liquid handling workstations (Hamilton Company, Reno, NV USA), and then sequenced on NextSeq sequencers (Illumina, Inc, San Diego, CA USA)as described in their publications [9]. The results are available once ready in the Karius portal, and clinical microbiology personnel routinely log in to the system, download the results, and upload them to our electronic medical records (EMRs).

All adult patients with KT results from UC San Diego Health (UCSD Health) from July 2017 to April 2024 were included. The patients’ medical records were reviewed to determine whether the KT had identified a pathogen and changed patient care. Basic demographics, turnaround time (defined as the time from the specimen being received by the central processing laboratory to the result uploaded to the EMR), test results, reason for testing, and patient outcome were collected to assess the clinical utility of the KT in the adult population.

For each case, we implemented criteria (Table 1) to determine the KT’s clinical utility. Clinical utility was calculated by the number needed to test (NNT). This review was determined to be IRB-exempt from the UCSD Health Aligning and Coordinating Quality Improvement, Research, and Evaluation (ACQUIRE) Committee.

## 3. Results

A total of 110 KT requests yielded 88 KTs from 88 unique patients after excluding duplicates, cancelations, or newborn patients. The median age of 88 patients included in this cohort was 58 years old (SD 17.5, from 18 to 88 years old; 1st quartile: 40 years old, 3rd quartile: 68 years old). Thirty-four (38.6%) tests were from female patients, and about half of the patients were white (Table 2). Of the 88 tests, 49 (55.7%) identified at least one pathogen. The median number of hospitalized days from admission to test ordering was 8.5 days. The median test turnaround time was three days (1st quartile: 2 days, 3rd quartile: 5 days).

In evaluating indication for testing, 25% (n = 22) tests were obtained for prolonged fever or sepsis of unknown origin, 25% (n = 22) with culture-negative endocarditis signs or symptoms, 6.8% (n = 6) for vascular graft infection, 21.6% (n = 19) for central nervous system pathology including clinical meningoencephalitis, brain abscess or lesion, vasculitis, mycotic aneurysm, or endophthalmitis, 11.4% (n = 10) for lung pathology including nodules, organizing pneumonia or pneumonitis, pleural effusion, cavitary lesion, or acute respiratory distress syndrome, 6.8% (n = 6) for osteomyelitis, discitis, or epidural abscess, and 2.3% (n = 2) for culture-negative septic joint (Table 3).

Among the eleven cases with a positive clinical impact, the KT was the only method to establish a diagnosis in eight cases, and the KT was faster than conventional microbiology testing in three cases. A complete list of organisms identified on the KT led to positive clinical impacts in brief clinical scenarios, as shown in Table 4. Among clinical syndromes, the vascular or graft infections category showed promising results (4 out of 6 cases). Among 56 cases with no clinical effect, 50% (n = 28) of cases were negative, which means no cfDNA of any microorganisms in the Karius database was detected in statistically significant amounts. 23.9% (n = 21) cases were deemed to have neutral clinical impacts. The number needed to test was 6.1; if all neutral results were also considered impactful, then the number needed to test was 2.75.

## 4. Discussion

Diagnosing infectious diseases remains challenging and often impossible without an invasive diagnostic procedure, resulting in less effective empiric treatment, prolonged hospital stays, and complications from hospitalization and invasive procedures. In this single tertiary academic center retrospective study, we performed the retrospective clinical utilization analysis of the KT.

A few studies have investigated the KT’s performance characteristics and clinical utility on serum samples among the general patient population. The clinical impact of the KT varies significantly, ranging from 7% to 56% depending on the patient population and/or clinical syndrome studied [10]. For example, in a single-center retrospective study, the KT was found to be compatible with the definitive diagnosis in 65% of cases and was assessed as having a positive impact in 43% (n = 34) of cases [11]. The KT was associated with superior clinical utility in both solid organ transplant recipients and sepsis patients. In another single-center retrospective review, the KT was assessed among immunocompromised patients, including those with B-cell and T-cell deficits. 58% of tests in this cohort identified at least one organism, and a positive test resulted in a modification to therapy in 14 of 21 patients [12]. Recently, another single-center study showed that approximately 46% (n = 59) of tests yielded a clinically significant pathogen [13]. On the contrary, the positive clinical impact rate in another multicenter study was only 7.3% (n = 6), while 86.6% (n = 71) of cases were determined to have no clinical impact [14]. Another study using the standardized criteria from this study assessed 80 patients without predefined clinical syndromes and found the rate of positive clinical impact to be 43% (n = 34), with a considerable 55% (n = 44) of cases having no clinical impact [15].

Our study implemented criteria developed and validated by two separate infectious disease specialists (SJ and AK), with an independent review process. We applied strict rules when defining the cases with positive impact, which led to a 12.5% positive rate (11 out of 88 cases). Defining cases with neutral impact was the weakest point of this chart review, as it could be provider dependent. For example, both discontinuing empiric coverage after the negative result and using the test to exclude other causes of treatment failure or clinical worsening fell into this category. In some cases, the KT was ordered in conjunction with other conventional microbiology diagnostic tests, but the results of the KT were obtained later than those of traditional testing.

Due to the high cost of testing, currently $2200 per test, we were curious about how these tests are reimbursed. We inquired with our billing department to gain clarity. Since most of our tests were ordered in an inpatient setting, we were able to audit a few patients’ revenue during their hospital stays. However, for most patients, payment for the total cost of the hospital stay is not separated into individual test reimbursements (as seen in Medicare’s Diagnosis-Related Groups (DRGs)); it was not possible to determine the reimbursement explicitly allocated to this test. Outpatient cases, where these tests may be reimbursed individually, represent a minority of the patients tested. In both scenarios, the cost of testing represents a significant financial burden on the healthcare system and patients. Therefore, we recommend regulating the test rather than allowing free access to ordering as part of a stewardship effort, primarily led by infectious disease specialists.

There are important limitations to these findings. This retrospective review included patients from a single healthcare system in Southern California with mainly adult patients, potentially limiting the generalizability of the findings. In our system, due to test costs, the KT is often sent late in a patient’s clinical course, which biases conventional diagnostic testing to result faster, even if the KT made the same diagnosis. Specimens may also be held before a decision is made to send the test to Karius, which increases the turnaround time for the measured result. Lastly, this study did not investigate when the specimens were collected relative to prior antibiotic therapy, which may affect test performance. The KT is also limited to cfDNA and publishes a finite but extensive list of reportable pathogens. This means that if the cohort of patients was impacted by RNA viruses and/or pathogens not on the published reportable pathogen list, it could bias against the impact. Lastly, Karius does not share individual sequence data with customers, which could aid in tracking and investigating nosocomial infection outbreaks.

## 5. Conclusions

In our inpatient setting, the KT had a positive clinical impact in 12.5% of cases, with a number needed to test of 6.1, and showed the highest yield in vascular graft infections. Negative results were also clinically useful in ruling out infection. Given the high cost of testing ($2200 per assay), lack of U.S. Food and Drug Administration approval, and potential out-of-pocket burden to patients, its use should be targeted to scenarios with a high pre-test probability and managed through stewardship by infectious disease and microbiology specialists. Cost reduction could further improve its role as a complementary diagnostic tool in select cases.

## Figures and Tables

**Table 1 pathogens-14-00811-t001:** Criteria to Determine Clinical Impact of Karius cfDNA NGS Test.

Criteria	Description
Positive Impact	Faster/Only diagnosis; clinicians clearly acted on result
Neutral Impact	Helped rule out infectious causes with modification to the treatment (e.g., discontinuation of empiric coverage) or strengthen the existing diagnosis (subjective decision by the ordering providers or reviewers)
Negative Impact	Either diagnosed existing infection or did not modify treatment

**Table 2 pathogens-14-00811-t002:** Demographic Characteristics.

Age (Years)	58 ± 17.5 (Median ± SD)
Sex (n, %)	
Male	53 (60.2%)
Female	34 (38.7%)
Transgender	1 (1.1%)
Total	88
Race/Ethnicity (n, %)	
White	44 (50.0%)
Hispanic	21 (23.9%)
African American	5 (5.7%)
Asian	12 (13.6%)
Other	6 (6.8%)
Total	88

**Table 3 pathogens-14-00811-t003:** Impact of Karius cfDNA NGS Test by Test Indications.

Indication of Testing	Total (N)	Positive Impact	Neutral Impact	No Impact
Fever, Sepsis	22	2	10	10
Culture-Negative Endocarditis	22	1	5	16
Vascular Graft Infection	6	4	0	2
Meningoencephalitis	10	1	3	6
Brain Abscess or Lesion	5	0	1	4
Endophthalmitis	1	1	0	0
Central Nervous System Vasculitis or Mycotic Aneurysm	4	1	1	2
Lung Pathology	10	1	3	6
Osteomyelitis, Discitis, or Epidural Abscess	6	0	1	5
Culture-Negative Septic Joint	2	0	0	2

**Table 4 pathogens-14-00811-t004:** Complete List of Organisms and Clinical Scenarios Identified on Karius cfDNA NGS Led to Positive Impacts.

Organism	Clinical Scenarios
*Aspergillus fumigatus*	Endophthalmitis
*Herpes Simplex Virus 1*	Encephalitis
*Pseudomonas aeruginosa*	Organizing Pneumonia
*Escherichia coli*	Culture-Negative Endocarditis
*Staphylococcus epidermidis*	Aortic Graft Infection
*Fusobacterium*	Aortic Graft Infection
*Rothia dentocariosa*	Iliac Artery Pseudoaneurysm
*Stenotrophomonas maltophilia*	Aorto-Bifemoral Bypass Graft Infection
*Rhizomucor pusillus*	Disseminated in Acute Myeloid Leukemia Patient
*Scedosporium boydii*	Posterior Inferior Cerebellar Artery Mycotic Aneurysm in Liver Transplant Recipient
*Cytomegalovirus*	Septic Shock in Liver Transplant Candidate

## Data Availability

The original contributions presented in this study are included in the article. Further inquiries can be directed to the corresponding author(s).

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
