# Peer review of "A Cell-Free DNA Plasma Next-Generation Sequencing Test—Is It Worth the Cost?"

_pathogens, 2025, doi:10.3390/pathogens14080811_

Round 1
Reviewer 1 Report
Comments and Suggestions for Authors
The authors present a succinct study in which the potential of cell free DNA samples being used as a diagnostic tool for infectious diseases is examined. Briefly, the authors review a number of medical cases in which Karius tests were employed as a diagnostic approach. These cases were reviewed and assessed based on the impact and efficacy of the Karius test in each respective context, with the authors determining the approach does have potential, but perhaps currently within certain divisions of infectious disease.
In reviewing the manuscript I made a couple of observations. The following should be considered by the authors when preparing a suitable revision.
- In the methods section, it is said that ‘next generation sequencing libraries are prepared and performed as previously described’-more detail should be given on this without alluding to other sources of information.
- When the records were examined, how many cases were available? What proportion of the total number does 88 cases that were examined represent?
- In Table 2, there are 61 males, 36 females, and 1 transgender comprising the case load reviewed for this study. This is indicated to be 88 cases, but should it not be 98?
- Similarly, when the ethnicity is counted up, it adds up to 100 cases. This needs to be checked.
Reviewer 2 Report
Comments and Suggestions for Authors
Thank you for the opportunity to review this manuscript. The topic is relevant, but several aspects need clarification or improvement:
In methods section:
- Please specify whether ethical approval and patient consent for data use were obtained.
- Some paragraphs, particularly in the Methods section, would benefit from a more professional and scientific tone.
- More clinically relevant details about the KT test should be included — for example, the time to result and how this may impact clinical decision-making.
Results: If race/ethnicity was not analyzed as a variable, clarify why it was reported.
Discussion: The section on cost should be simplified or made clearer.
Expand on the limitations of both the study and the KT itself. Also, discuss potential usefulness in nosocomial invasive infections, even hypothetically.
Reviewer 3 Report
Comments and Suggestions for Authors
The manuscript entitled "Cell-free DNA plasma Next-Generation Sequencing Test - Is Worth the Cost?" written by Jung et al. is very interesting and I think it has the potential to fill a gap in literature. As diagnostic methods are evolving and newer methods are being described, I think it is very important to evaluate their efficiency and the way by which they can be implemented in current practice. While I do believe this manuscript describes a very important topic, I think there are several areas in which it can be improved. I have made some suggestions below:
- in line 63, the abbreviations KT is explained. However, the abbreviation has already been used several times in the text. Can you please add the full-word followed by the abbreviation the first time they are mentioned in the manuscript?
- I think the authors should add a paragraph in the introduction section highlighting the exact aim of the manuscript more, underlining why they believe the study is important and what gap in literature they are trying to fill. With the way the aim is currently written, it's not very clear what their manuscript is trying to do
- lines 69-70 - where was the method previously described? Nothing is mentioned about it in the article so far and no citation to another study is available either. Even if the authors used a previously described method, I think it should still be briefly described so that is more easily accesible to the reader
- nothing is mentioned in the material and methods section regarding data analysis. Please revise
- Table 4 is not really a table, just a column with some associations that are sometimes not even written in a consistent way. I think those results would be more suitable to be presented either as a table with 2 columns (which I think was the initial intention of the authors) or as a list
- in the results section, I think you could also add some graphic representation of the data to better present the results and make them more easily accesible to the reader
- line 127 - when writing the number of cases in brackets, please use "n=". Please replace "43% (34)" with "43% (n=34)"
- in the last paragraph of the discussion section, nothing is mentioned regarding the limitations of the study. Please revise
- the conclusion paragraph should be its own section. Please correct
Round 2
Reviewer 1 Report
Comments and Suggestions for Authors
The authors have addressed my comments.
Author Response
We are grateful to the Editor and Reviewers for their valuable feedback, insightful observations, and constructive recommendations, which have greatly improved the clarity, depth, and overall quality of our work. The comments provided were instrumental in refining our approach and strengthening our lines of investigation. We trust that the revisions undertaken in response to your suggestions have further enhanced the rigor and impact of our manuscript.
Reviewer 2 Report
Comments and Suggestions for Authors
The conclusion should be reformulated to clearly reflect and be directly supported by the findings of the present study.
Author Response
Comments 1:
The conclusion should be reformulated to clearly reflect and be directly supported by the findings of the present study.
Response 1:
In our inpatient setting, the KT had a positive clinical impact in 12.5% of cases, with a number needed to test of 6.1, and showed the highest yield in vascular graft infections. Negative results were also clinically useful in ruling out infection. Given the high cost of testing ($2,200 per assay), lack of U.S. Food and Drug Administration approval, and potential out-of-pocket burden to patients, its use should be targeted to scenarios with a high pre-test probability and managed through stewardship by infectious disease and microbiology specialists. Cost reduction could further improve its role as a complementary diagnostic tool in select cases.
Reviewer 3 Report
Comments and Suggestions for Authors
I would like to thank the authors for taking into consideration my suggestions and revising the manuscript. The article has been improved and can be accepted in present form.
Author Response

(The authors gave the same response as above.)
